# Impact of patients' age and comorbidities on prostate cancer overdiagnosis in clinical practice

Abraham Beltrán[1], Lucy Parker[1,2], Irene Moral-Pérez[3], Juan Pablo Caballero-Romeu[4], Elisa Chilet-Rosell[1,2], Ildefonso Hernández-Aguado[1,2], Pablo Alonso-Coello[2,5], Elena Ronda[2,6], Luis Gómez-Pérez[7], Blanca Lumbreras[1,2]*

1 Department of Public Health, History of Science and Gynecology, Miguel Hernandez University, San Juan de Alicante, Spain, 2 CIBER of Epidemiology and Public Health, CIBERESP, Madrid, Spain, 3 EAP Sardenya, Institut de Recerca Sant Pau, Barcelona, Spain, 4 Department of Urology, Dr. Balmis General University Hospital, Alicante Institute for Health and Biomedical Research (ISABIAL), Alicante, Spain, 5 Iberoamerican Cochrane Centre, Department of Clinical Epidemiology and Public Health, Biomedical Research Institute Sant Pau, Barcelona, Spain, 6 Public Health Research group, Alicante University, San Vicente del Raspeig, Spain, 7 Department of Urology, University General Hospital of Elche, Elche, Spain

* blumbreras@umh.es

**Data Availability Statement:** All data files are available from the Zenodo database (doi:10.5281/zenodo.14509461).

## Abstract

### Introduction

Overdiagnosis in PSA-based prostate cancer (PCa) screening is primarily studied in younger, healthier populations from clinical trials. This study aimed to evaluate the probability of overdiagnosis in PCa screening within a clinical practice context, focusing on its relationship with PSA levels, Gleason scores, and subsequent clinical procedures.

### Methods

We conducted a retrospective cohort analysis of 1,070 asymptomatic men over 40 years old diagnosed with PCa between 2004 and 2022, following a positive PSA test. The patients were followed until December 31, 2022, with a median follow-up time of 5.7 years (IQR 3.2–8.6). The primary outcome was the probability of overdiagnosis, assessed through life expectancy and the Charlson Comorbidity Index, considering lead times of 5, 10, and 15 years.

### Results

We found that patients with PSA levels >10 ng/dL and/or Gleason scores ≥8 were generally older and had more comorbidities than those with PSA levels 4–10 ng/dL and/or Gleason scores ≤7. The probability of overdiagnosis was significantly higher in patients with PSA levels >10 ng/dL (41.4%, IQR 21.5–73.9) and Gleason scores ≥8 (42.6%, IQR 14.9–38.9), compared to those with PSA levels 4–10 ng/dL (20.1%, IQR 12.8–30.4) and Gleason scores ≤7 (26.6%, IQR 23.6–68.6). Notably, 71.7% of patients did not receive pharmacological treatment. Patients with higher PSA levels also experienced greater radiation exposure from diagnostic imaging (median 19.9 mSv vs. 14.7 mSv, p = 0.004).

**Funding:** Research was funded by the research project of the Instituto de Salud Carlos III, code PI20/01334, Principal Investigator Dr. Blanca Lumbreras, co-financed with FEDER funds from 631 the European Union "A way of doing Europe". The funders had no role in study design, data collection and analysis, decision to publish, or preparation of the manuscript.

**Competing interests:** The authors have declared that no competing interests exist.

## Conclusions

These findings underscore the high likelihood of overdiagnosis in older patients with elevated PSA levels and significant comorbidities, highlighting the need for careful consideration of patient comorbidities before PSA testing.

## Introduction

Controversy surrounding the use of the prostate-specific antigen (PSA) test as a Prostate Cancer (PCa) screening tool is mainly related to the probability of overdiagnosis and overtreatment of indolent cancers [1]. Overdiagnosis is defined as individuals receiving a diagnosis with a condition that would never have become symptomatic before the end of the individual's life [2].

The probability of overdiagnosis in PCa screening with the PSA test has been described as ranging from 27% to 56% (depending on the screening protocol) [3, 4]. However, there is no consensus regarding how best to measure the probability of overdiagnosis in practice due to its complexity. According to evidence [5], there are four different approaches (follow-up of randomized controlled trials; studies based on histopathology and imaging procedures; modelling studies, and epidemiological cohort studies), all of which have both advantages and disadvantages. A previous systematic review of autopsy studies [6] showed a high prevalence of PCa as incidental findings, although the high variability in the pathology assessments of the included studies affected the validity of this review. The European Randomized Study of Screening for Prostate Cancer (ERSPC) [7] estimated a 41% probability of overdiagnosis, but this was based on modelling approaches rather than calculations from the observed data. However, other major clinical trials such as the Prostate, Lung, Colorectal and Ovarian (PLCO) Cancer Screening Trial [8], did not evaluate the probability of overdiagnosis. Nevertheless, the population identified through randomized studies and those included in modelling approaches differ from those who undergo a PSA test in clinical practice. Typically, the latter group tends to be older and presents a higher prevalence of comorbidities, thereby decreasing the patient's life expectancy and potentially increasing the likelihood of overdiagnosis.

As such, there is limited evidence regarding the likelihood of overdiagnosis in men who undergo a PSA test as part of opportunistic screening in a clinical practice setting, which can be estimated using real-world data. In addition, the available studies included the patients' age in the estimation of their life expectancy, but their comorbidities are not usually considered [9]. A recent update of the EAU-EANM-ESTRO-ESUR-ISUP-SIOG Guidelines on Prostate Cancer [10] includes an algorithm aims at reducing the probability of overdiagnosis, which emphasizes the need to consider patients' comorbidities when assessing their PCa risk. Therefore, it is essential to estimate the probability of probability of overdiagnosis in clinical practice, taking comorbidities into account in the analysis. These data could support clinicians when they decide to order a new PSA test.

Moreover, the variables related to a higher probability of overdiagnosis could be different in clinical practice from those in clinical trials. Previous research, for instance, found that the presence of overdiagnosis was more likely in older patients and in those with lower level of PSA test [11]. However, in contrast, it has been described that high levels of PSA test might be associated with the presence of different types of cancer and noncancer diseases [12], hence, with lower life expectancy and higher probability of overdiagnosis.

The adverse consequences of overdiagnosis include negative effects on patient health, mainly related to overtreatment and excess clinical interventions [13]. For instance, the

excessive utilization of imaging tests during patient follow-up implies radiation exposure, linked to stochastic health effects such as cancer [14]. Nevertheless, there is no evidence regarding the excess of radiation exposure during the management of patients diagnosed with PCa through opportunistic screening with the PSA test, nor on the pharmacological treatment administered.

In this study, we aimed to assess the probability of overdiagnosis by considering patients' age and comorbidities in the estimation of their life expectancy and examining its relationship with PSA levels and Gleason scores through a retrospective cohort analysis in clinical practice. Additionally, we investigated the pharmacological treatments and radiation exposure resulting from diagnostic imaging tests during patients' follow-up.

## Material and methods

### Design

We conducted a retrospective cohort study to assess the probability of overdiagnosis in PCa opportunistic screening at two Spanish hospitals between 2004 and 2022. The patient's entry in the study was the date of PCa diagnosis, and the endpoint was the end of follow-up (either 31st December 2022, or the date of death, whichever came first).

### Setting

Two departments of health in the Valencian Community, Southeast of Spain: Health Department number 19 (Catchment area of the General University Hospital of Alicante, 255,439 inhabitants) and Health Department number 17 (Catchment area of the General University Hospital of S. Joan d'Alacant, 233,115 inhabitants).

### Subjects

We initially set up a cohort of all men >40 years old who were registered in the Valencian Health Departments of the two included hospitals in 2004 through their Population Information System (SIP), and who were diagnosed with PCa during the time for study (2004–2022). For this analysis, we included patients with a positive PSA test result at least 12 months before the diagnosis of PCa (likely indicating PSA tests that led directly to the diagnosis). We excluded patients with a previous PCa diagnosis and those with prostatic symptoms at the time of PSA testing. We accessed this data in March 2023.

### Sample size estimation

To estimate the frequency of overdiagnosis, we assumed an expected frequency between 40 and 64% depending on the lead time considered (assuming lead-times of 5, 10 and 15 years, as previously done [15], we were able to detect 40–64% of PSA-detected cases as overdiagnosis with a 3% precision (95% CI). The frequency of PCa in our setting is 70.8/100,000 according to Spanish data [16]. Considering that in the two health departments involved there are 134,355 men over 40 years of age, more than 1,700 new cases of PCa can be included in the study and thus achieve a precision of the estimates of 3%.

### Variables

The main outcome variable was potential overdiagnosis (estimated considering a lead time of 5, 10 or 15 years) defined as a diagnosis of PCa received by an asymptomatic individual through PSA screening, who would have died from a cause unrelated to PCa within a certain period of time, and during which time they would not have experienced effects related to their PCa.

Recent biological and epidemiological studies on certain types of tumors such as breast cancer or PCa have shown that the exponential function fits well with the sojourn time (period during a tumor is asymptomatic but screen-detectable) [17]. In PCa detected by PSA, the probability that the negative effects of PCa would have taken longer to appear than the estimated life expectancy within a predefined period can be calculated as exp(-t/M), where t is the estimate of life expectancy and M is the lead time estimates from the literature for screen detected prostate cancer [15].

Firstly, we estimated the life expectancy for each patient (defined as the real survival probability for each patient) considering the theoretical survival probability for each patient according to the Spanish male population's life expectancy in the year 2022 provided by the National Institute of Statistics, adjusted by the patient's comorbidities [18] according to the following formula: $x = p^y$,

where:

x = Real survival probability for each patient

p = Theorical survival probability for each patient as his/her age when healthy, given by the National Institute of Statistics

$y = e^{0.9*comorbidity}$

comorbidity = Total score of comorbidities for each patient

To obtain the total score of comorbidities for each patient we used the Charlson Comorbidity Index, which consists of 17 items based on the International Classification of Diseases (ICD) 9 and 10 codes [19]. The Charlson Comorbidity Index is a method of categorizing comorbidities of patients based on the ICD diagnosis codes found in administrative data, such as hospital abstracts data. Each comorbidity category has an associated weight (from 1 to 6), based on the adjusted risk of mortality or resource use, and the sum of all the weights results in a single comorbidity score for a patient. A score of zero indicates that no comorbidities were found. The higher the score, the more likely the predicted outcome will result in mortality or higher resource use. We obtained patients' comorbidities through medical records.

We calculated the probability of overdiagnosis of each patient for lead times 5, 10 and 15 years.

We assessed the pharmacological treatment received by the patients during follow-up and classified it as chemotherapy, androgen antagonists, both treatments or no treatment. We also evaluated the exposure to radiation (mSv) due to diagnostic imaging tests during the period of study and estimated the cumulative effective dose in the study population according to previous evidence [20].

Independent variables:

- Gleason score: The International Society of Urological Pathology 2005 Gleason score together with its 2014 and 2019 modifications was used for PCa grading system [21, 22].

- PSA level: According to the latest recommendations from the European Association of Urology [10], a positive PSA result is defined if the total PSA value is >10 ng/ml, and in those cases where the total PSA values are between 4 and 10 ng/ml if the value of the free PSA/total PSA fraction is < 25% (in at least two determinations).

## Statistical analysis

Firstly, the characteristics of the participants were described stratifying by presence or absence of symptoms to compare both groups. Differences in the probability of overdiagnosis and life

expectancy according to PSA levels and Gleason score were assessed. The primary analysis evaluated the probability of overdiagnosis according to PSA level (4–10 ng/dl and >10 ng/dl) and Gleason score ($\leq$ 6, 7 and $\geq$8).

The number of valid cases and percentages, the median and the interquartile range were used to summarize available data. Comparisons were carried out using the Wilcoxon rank sum test or the chi-square test, depending on the characteristics of the variables.

The comparison between individuals who died from prostate cancer and those who died from other causes was visualized using the survival function, which represents the probability of the event occurring at time t. To create the survival graph, we used the geom_step function from the ggplot2 library.

The analysis was carried out using RStudio Team (2022). RStudio: Integrated Development Environment for R (Version 4.2.2).

### Ethic statement

CEIC Sant Joan d'Alacant (20/041) on 8 January 2021 (consent from the participants was waived). the study was performed in accordance with the Declaration of Helsinki.

## Results

### Description of the subjects included in the study

Out of the 2,331 patients > 40 years diagnosed with PCa in the years 2004–2022, 1,070 asymptomatic patients with a positive PSA test were included in the study (Fig 1). The median age of these patients was 70 years old (IQR 64–77 years) and the median time of follow-up was 5.7 years (IQR 3.2–8.6 years). Of the 1,070 men, 530 patients had a PSA 4–10 mg/ml and 540 a PSA >10 mg/ml. Of the 464 patients who had a Gleason score, 164 (35.3%) had a Gleason score 7, and 101 (21.8%) had a Gleason score $\geq$8.

Median life expectancy at diagnosis was 13 years (IQR 6–19 years). Most of the patients were alive at the end of the study (921, 86.1%), 124 (11.6%) died from other causes different from PCa and 25 (2.3%) died from PCa.

### Life expectancy and probability of overdiagnosis

We estimated the probability of overdiagnosis corresponding to each lead time (5, 10 and 15 years) for patients with life expectancy < = 10 years and for those with life expectancy > 10 years (Fig 2).

Of 1,070 patients, 393 (36.7%) had an expectancy of life $\leq$10 years and 677 (63.3) > 10 years. The median probability of overdiagnosis considering a lead time of 5 years was 7.1% (IQR 2.2–31.4): for those patients with a life expectancy $\leq$10 years, it was 46.32% (IQR 28–100) and for those with a life expectancy >10 years, it was 3% (IQR 1.4–6.2) (p<0.001). The median probability of overdiagnosis considering a lead time of 10 years was 26.6% (IQR 14.9–56.1): for those patients with a life expectancy $\leq$10 years, it was 68.1% (IQR 52.9–100) and for those with a life expectancy >10 years, it was 17.3% (IQR 11.8–24.8) (p<0.001). The median probability of overdiagnosis considering a lead time of 15 years was 41.3% (28.1–67.9): for those patients with a life expectancy $\leq$10 years, it was 77.4% (IQR 65.4–100) and for those with a life expectancy >10 years, it was 31.1% (IQR 24.1–39.5) (p<0.001).

### Life expectancy at the time of PCa diagnosis and probability of overdiagnosis according to PSA level and Gleason score

Table 1 shows that patients with PSA levels >10 ng/dl were more likely to have lower life expectancy at PCa diagnosis than those with PSA levels between 4–10 ng/dl (median 8.8 years,

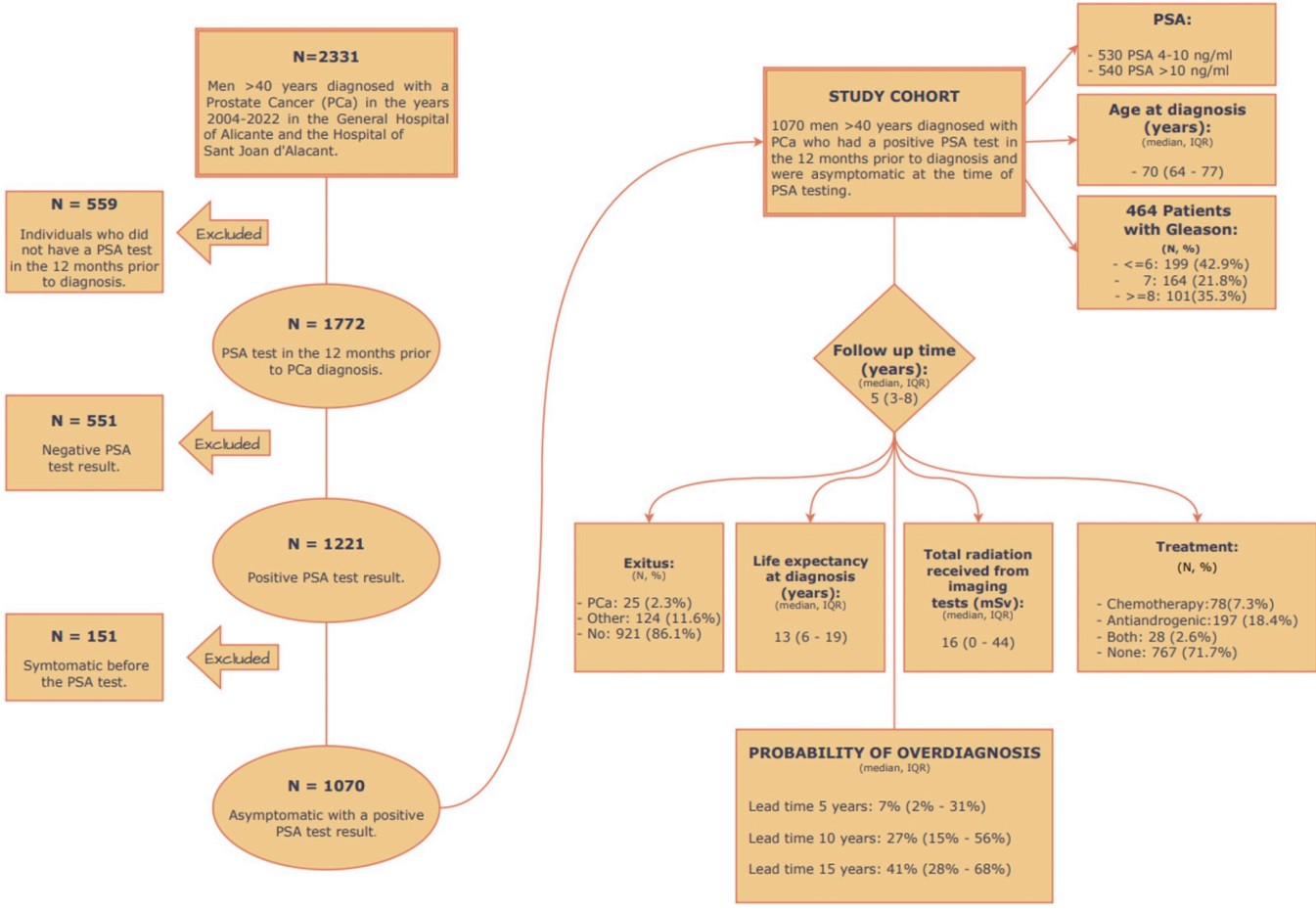

**Fig 1. Description of the flow diagram of the patients finally included in the study.**

IQR 3.0–15.4 years vs median 16.1 years, IQR 11.9–20.6 years, p<0.001). In addition, patients with PSA levels >10 ng/dl were more likely to have greater probability of overdiagnosis than those with PSA levels between 4–10 ng/dl (p<0.001 in all the comparisons) with all 3 lead-time estimations considered (5, 10 and 15 years).

Patients with a Gleason score ≥8 tended to have lower life expectancy at PCa diagnosis (median 8.5 years, IQR 3.8–14.5 years) than those with a Gleason score 7 (median 13.3 years, IQR 9.4–19 years) and those with a Gleason score ≤6 (median 16.1 years, IQR 11.9–20.6 years), p<0.001). In addition, patients with a Gleason score ≥8 were more likely to have a higher probability of overdiagnosis than those with a Gleason score 7 and those with a Gleason score ≤6 (p<0.001 in all the comparisons).

Patients with PSA levels 4–10 ng/dl were less likely to have comorbidities than patients with PSA levels >10 ng/dl (419, 79.1% vs 313, 58%, p<0.001). In addition, patients with PSA>10 ng/dl were more likely to be older than patients with PSA levels 4–10 ng/dl (median age 73 years, IQR 67–82 vs median age 67 years, IQR 62–72, p<0.001). Patients with Gleason score ≤6 and 7 showed a lower probability to have comorbidities than patients with Gleason score ≥8 (142, 71.4%, 117, 71.3%, and 53, 52.5%, respectively, p<0.001). In addition, patients with Gleason score ≥8 were more likely to be older than patients with Gleason score ≤6 and 7 (median age 73 years, IQR 67–87, median age 66, IQR 61–71, and median age 69 years, IQR 63–76, respectively, p<0.001) (Table 2).

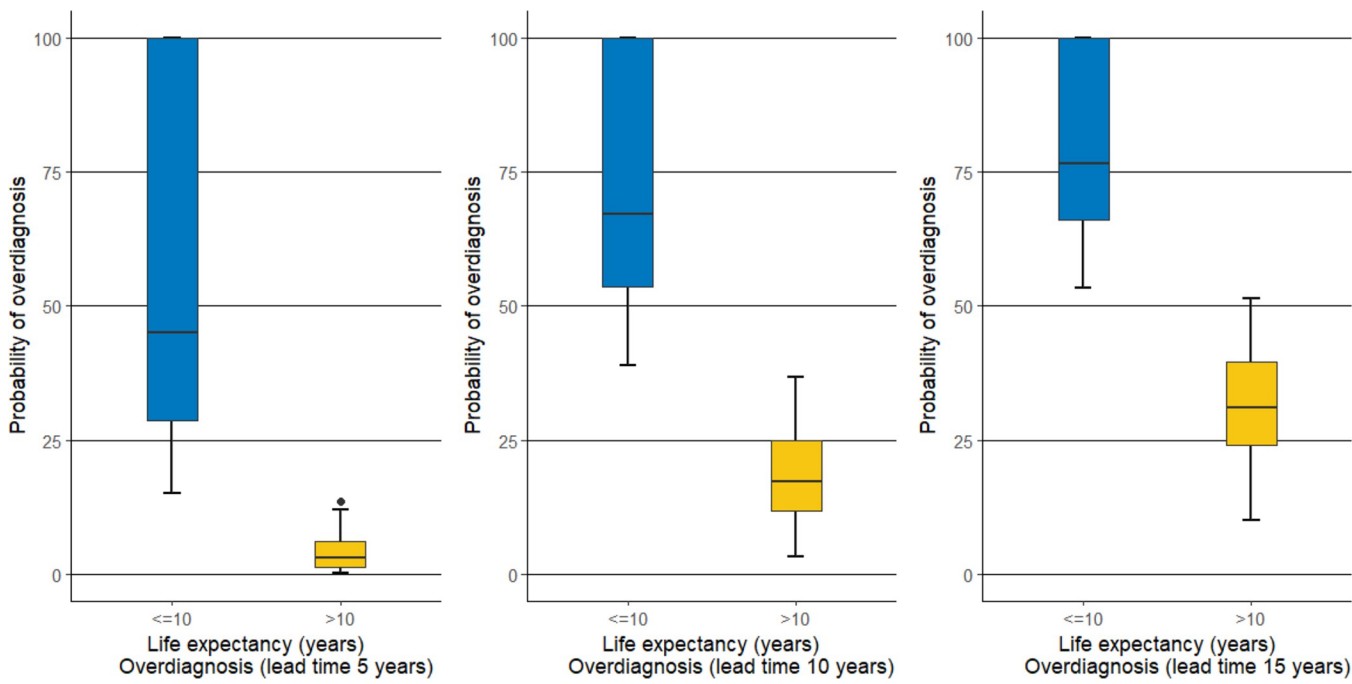

**Fig 2. Comparison of the probability of overdiagnosis based on lead times of 5, 10 and 15 years according to the patients' life expectancy higher or lower than 10 years.**

## Analysis of the mortality from PCa and from other causes

Overall, 149 (13.9%) patients died during the followed-up period, 25 (2.3%) died from PCa and 124 (11.6%) due from other causes different from PCa. There were no differences in the time of follow-up between patients who died from PCa (median 4.3 years, IQR 1.9–8.1), those who died from other causes (median 5.1 years, IQR 2.1–8.1) and those who survived (median 5.2 years, IQR 3.1–8.2), p = 0.173.

Fig 3 shows the mortality rate/1,000 persons adjusted by follow-up time from PCa and from other causes according to PSA level (Fig 3A) and Gleason score (Fig 3B). Patients with PSA levels ≥ 10 ng/dl were more likely to die than patients with PSA levels between 4–10 mg

**Table 1. Life expectancy at the time of diagnosis and probability of overdiagnosis at 5, 10, and 15 years, according to PSA level and Gleason score only in asymptomatic patients with a previous positive PSA value 12 months before PCa diagnosis.**

| Variables | N (%) | Life expectancy at diagnosis (years) | | | Probability of overdiagnosis (lead time 5 years) (%) | | | Probability of overdiagnosis (lead time 10 years) (%) | | | Probability of overdiagnosis (lead time 15 years) (%) | | |
|---|---|---|---|---|---|---|---|---|---|---|---|---|---|
| | | Median | IQR | *p-value* | Median | IQR | *p-value* | Median | IQR | *p-value* | Median | IQR | *p-value* |
| **PSA\* (ng/ml)** | **1070** | **13** | **6–19** | *<0.001* | **7** | **2–31** | *<0.001* | **27** | **15–56** | *<0.001* | **41** | **28–68** | *<0.001* |
| **4–10** | 530 (49.5%) | 16.1 | 11.9–20.6 | | 4.0 | 1.6–9.2 | | 20.1 | 12.8–30.4 | | 34.3 | 25.4–45.2 | |
| **>10** | 540 (50.5%) | 8.8 | 3.0–15.4 | | 17.2 | 4.6–54.6 | | 41.4 | 21.5–73.9 | | 55.5 | 35.9–81.7 | |
| **Gleason score** | **464** | **14** | **8–20** | *<0.001* | **6.2** | **1.9–19.8** | *<0.001* | **24.8** | **13.8–44.4** | *<0.001* | **39.5** | **26.7–58.2** | *<0.001* |
| **≤6** | 199 (42.9%) | 16.1 | 11.9–20.6 | | 4.0 | 1.6–9.2 | | 20.1 | 12.8–30.4 | | 34.3 | 25.4–45.2 | |
| **7** | 164 (35.3%) | 13.3 | 9.4–19.0 | | 7.1 | 2.2–15.2 | | 26.6 | 14.9–38.9 | | 41.3 | 28.1–53.3 | |
| **≥8** | 101 (21.8%) | 8.5 | 3.8–14.5 | | 18.2 | 5.6–47.0 | | 42.6 | 23.6–68.6 | | 56.6 | 38.1–77.8 | |

\* A positive PSA result is defined if the total PSA value is > 10 ng/ml, and in those cases where the total PSA values are between 4 and 10 ng/ml if the value of the free PSA/total PSA fraction is < 25% (at least in two determinations).

**Table 2. Description of the association between PSA levels and Gleason score and patient's comorbidities and age at PCa diagnosis.**

| | N (%) | Comorbidities | | | | | Age at PCa diagnosis (years) | | |
|---|---|---|---|---|---|---|---|---|---|
| | | **0** | **1–2** | **3–4** | **> = 5** | *p-value* | **Median** | **IQR** | *p-value* |
| **PSA (ng/ml)** | **(1070)** | **732 (68.4)** | **223 (20.8)** | **54 (5.1)** | **61 (5.7)** | <0.001 | **70** | **64–77** | <0.001 |
| 4–10 | 530 | 419 (79.1) | 90 (17.0) | 13 (2.4) | 8 (1.5) | | 67 | 62–72 | |
| >10 | 540 | 313 (58.0) | 133 (24.6) | 41 (7.6) | 53 (9.8) | | 73 | 67–82 | |
| **Gleason score** | **(464)** | **312 (67.2)** | **113 (24.4)** | **23 (5.0)** | **16 (3.4)** | 0.009 | **69** | **63–74** | <0.001 |
| ≤6 | 199 | 142 (71.4) | 44 (22.1) | 9 (4.5) | 4 (2.0) | | 66 | 61–71 | |
| 7 | 164 | 117 (71.3) | 37 (22.6) | 6 (3.7) | 4 (2.4) | | 69 | 63–76 | |
| ≥8 | 101 | 53 (52.5) | 32 (31.7) | 8 (7.9) | 8 (7.9) | | 73 | 67–77 | |

/dl. This was the case for non PCa mortality (mortality rate 176.3/1,000 persons vs mortality rate 54.7/1,000 persons) and PCa mortality (mortality rate 40.8/1,000 persons vs mortality rate 5.6/1,000 persons), p<0.001. Patients with Gleason score ≥8 were more likely to die from PCa than those with Gleason score 7 and Gleason score ≤6 (mortality rate 59.4/1,000 persons, 6.1/1,000 persons and 5/1,000 persons, respectively), p<0.001. We also observed large differences in mortality from other causes between patients with Gleason score ≥8 (mortality rate 158.4/1,000 persons) and those with Gleason score 7 (mortality rate 109.8/1,000 persons) and Gleason score ≤6 (mortality rate 100.5/1,000 persons), p<0.001.

Fig 4 represents the probability of survival at 12 years after PCa diagnosis. The probability of an individual dying from PCa in 12 years of follow-up was 12.3% (9.8% in patients with PSA levels 4–10 ng/dl and 10.7% in patients with PSA levels>10 ng/dl). The probability of dying from a cause other than PCa was four times more (22.5% in patients with PSA levels 4–10 ng/dl and 76.9% in patients with PSA levels>10 ng/dl).

## Description of the pharmacological treatment and exposure radiation from diagnostic imaging tests received by patients

Table 3 describes the pharmacological treatment and radiation from diagnostic imaging tests received by patients according to PSA level and Gleason score. The majority of patients did not receive any pharmacological treatment (767, 71.7%), with differences between patients with PSA levels between 4–10 ng/dl (408, 77%) and those with PSA levels > 10 ng/dl (359, 66.5%) (p<0.001). In addition, patients with PSA levels > 10 ng/dl were more likely to receive androgen antagonists (108, 20.0%) or chemotherapy (55, 10.2%) than patients with PSA levels 4–10 ng/dl (89, 16.8% and 23, 4.3%, respectively). However, there were no differences in the pharmacological treatment received according to the Gleason score.

**Table 3. Description of the pharmacological treatment and radiation from diagnostic imaging tests received by patients according to PSA level and Gleason score.**

| | N (%) | Pharmacological treatment | | | | | Exposure radiation (mSv) | | |
|---|---|---|---|---|---|---|---|---|---|
| | | **Androgen Antagonists** | **Chemotherapy** | **Both** | **None** | *p-value* | **Median** | **IQR** | *p-value* |
| **PSA (ng/ml)** | **(1070)** | **197 (18.4)** | **78 (7.3%)** | **28 (2.6%)** | **767 (71.7%)** | <0.001 | **16.2** | **0.6–44.0** | 0.004 |
| 4–10 | 530 (49.5) | 89 (16.8) | 23 (4.3) | 10 (1.9) | 408 (77.0) | | 14.7 | 0.3–38.4 | |
| >10 | 540 (50.5) | 108 (20.0) | 55 (10.2) | 18 (3.3) | 359 (66.5) | | 19.9 | 1.3–50.9 | |
| **Gleason score** | **(464)** | **108 (23.3)** | **39 (8.4)** | **15 (3.2)** | **302 (65.1)** | 0.070 | **31.5** | **16.3–62.6** | <0.001 |
| ≤6 | 199 (42.9) | 36 (18.1) | 13 (6.5) | 7 (3.5) | 143 (71.9) | | 25.1 | 8.1–55.3 | |
| 7 | 164 (35.3) | 45 (27.4) | 12 (7.3) | 5 (3.1) | 102 (62.2) | | 28.9 | 15.6–57.4 | |
| ≥8 | 101 (21.8) | 27 (26.7) | 14 (13.9) | 3 (3.0) | 57 (56.4) | | 42.9 | 22.6–77.3 | |

**A**

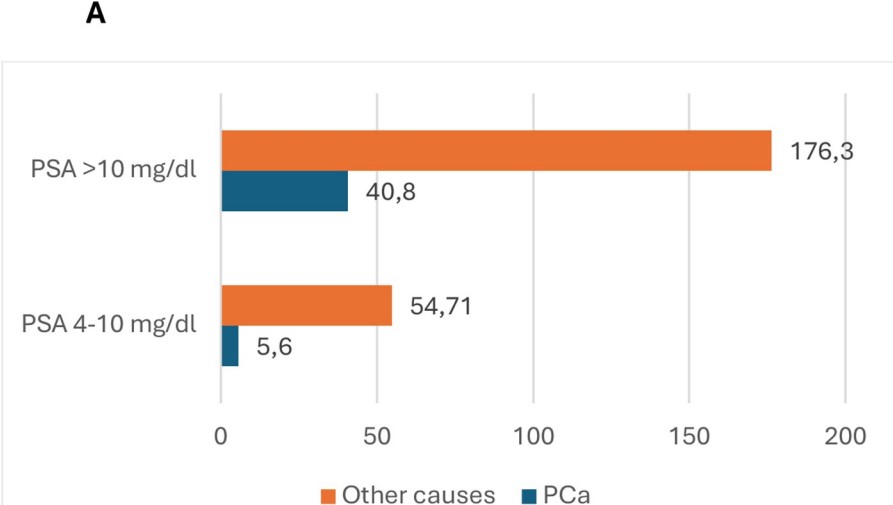

**B**

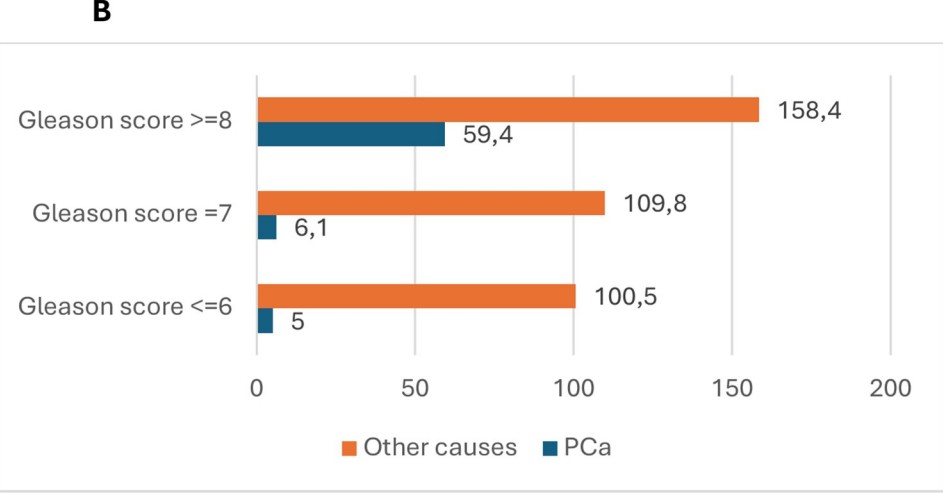

**Fig 3.** Mortality rate/1,000 persons from PCa and from other causes according to PSA level (Fig 3A) and Gleason score (Fig 3B) only in asymptomatic patients with a previous positive PSA value at the time of PCa diagnosis.

Median exposure radiation (mSv) received by the patients during the follow-up period was 16.2 mSv (IQR 0.6–44.0 mSv). Patients with PSA levels > 10 ng/dl were more likely to receive more radiation (median 19.9 mSv, IQR 1.3–50.9 mSv) than patients with PSA levels 4–10 ng/dl (median 14.7 mSv, IQR 0.3–38.4 mSv), p = 0.004. Patients with a Gleason score ≥ 8 tended to receive higher exposure to radiation (median 42.9 mSv, IQR 22.6–77.3 mSv) than those with a Gleason score 7 (median 28.9 mSv, IQR 15.6–57.4 mSv) and those with a Gleason score ≤6 (median 25.1 mSv, IQR 8.1–55.3 mSv), p<0.001.

## Discussion

We estimated that the median probability of overdiagnosis was 7% with a lead time of 5 years, 27% with a lead time of 10 years, and 41% with a lead time of 15 years. These estimates varied based on patients' life expectancy. Patients with a life expectancy of 10 years or less showed an overdiagnosis probability of over 50%. Additionally, patients with higher PSA levels and

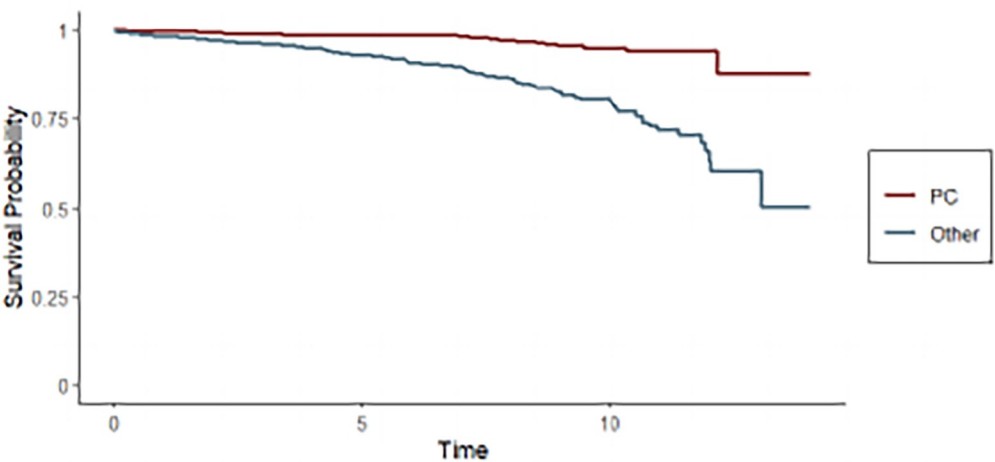

**Fig 4. Probability of survival at 12 years after prostate cancer diagnosis (comparing mortality from prostate cancer with all other causes) of all asymptomatic patients with a positive PSA at least 12 months before the diagnosis.**

Gleason scores greater than 8 were older, had more comorbidities, and consequently had shorter life expectancies and nearly double the overdiagnosis probability compared to those with lower PSA levels and/or Gleason scores of 7 or less. The probability of an individual dying from PCa within 12 years of follow-up was four times lower than the probability of dying from another cause. These probabilities also varied with PSA levels and Gleason scores. Furthermore, patients with higher PSA levels were more likely to receive pharmacological treatment and had greater exposure to radiation from diagnostic imaging tests compared to those with lower PSA levels.

Our estimate that 41% of screen-detected cancers were overdiagnosed (lead time 15 years) was similar to previously published estimates derived from clinical trials. A 16-year follow-up report from the European Randomised Study of Screening for Prostate Cancer (ERSPC) indicated that 43% of screen-detected cancers were overdiagnosed [23]. A recent review by the U.S. Preventive Services Task Force also noted that 20% to 50% of screen-detected prostate cancers were overdiagnosed [24]. However, patients in clinical trials, usually younger and with fewer comorbidities, should show a higher probability of overdiagnosis than those included in clinical practice, usually older and with more comorbidities. In fact, there were significant differences between patients included in our study in clinical practice and those included in previous clinical trials. In ERSPC, for example, patients were younger (median age 61 years, IQR 57.9–65) than those included in our study (median age 71 years, IQR 65–78) and generally with fewer comorbidities. The similar probability of overdiagnosis found in these two settings could be related to the population included in clinical practice being more heterogeneous than that included in clinical trials. In our study, almost half of the patients included had PSA levels >10 ng/dl and/or Gleason score ≥ 8 and, therefore, a lower probability of overdiagnosis (being older patients with more comorbidities), than the other half of the patients, with a higher probability of overdiagnosis. In clinical trials, the population is more homogeneous and most patients have a similar age and number of comorbidities, which leads to fewer differences in the probability of overdiagnosis between patients. Therefore, the assessment of life expectancy in clinical practice, considering not only the patient's age but also his comorbidities, is essential when the clinician wants to estimate the probability of overdiagnosis when ordering a PSA test.

A previous nomogram quantified the contribution of age, Gleason score, and PSA at diagnosis to the predicted likelihood of overdiagnosis [25]. After standard diagnostic assessment of model assumptions, these authors found that each additional year of age at diagnosis was associated with a 12.9% increase in the odds of overdiagnosis, showing that a Gleason score of 7 or higher was associated with a 19.5% decrease in the odds of overdiagnosis relative to Gleason score of 6 or lower and each additional 1 ng/dl of serum PSA up to 10 ng/dl was associated with a 16.6% decrease in the odds of overdiagnosis. Similarly, another study based on population-based incidence data from the US Surveillance, Epidemiology and End Results database concluded that overdiagnosis appeared to be predominately associated with age and low levels of PSA test [26]. In contrast, in our study, the odds of overdiagnosis increased in those patients with PSA > 10 ng/dl and those with Gleason score ≥8, because they were older patients with a higher number of comorbidities. Our results are in line with a previous population-based observational cohort (Prostate Cancer Database Sweden) [27] which also found that at PCa diagnosis, patients with more comorbidities (using as we did, the Charlson comorbidity index) tended to be older than those without any comorbidities, and the median PSA level at diagnosis was also higher in the comorbid group compared to the group with no comorbidities. Correspondingly, a greater percentage of low-grade, localized tumors were detected in the group of patients who did not have comorbidities then those who had them.

In relation to the different diagnostic pathway and treatments, patients with PSA levels >10ng/dl and those with Gleason score ≥8 were more likely to receive pharmacological treatment than patients with PSA levels between 4–10 ng/dl and those with lower Gleason scores. This is consistent with current guidelines in which active surveillance is the preferred treatment for low-risk CaP [10]. However, patients with elevated PSA test levels and Gleason score ≥8 were more likely to receive increased radiation exposure from imaging tests.

## Implications for clinical practice

Therefore, a detailed assessment of the prevalence of comorbidities among patients with PCa in real-life settings has significant implications for patient management. In line with the recent update of the EAU-EANM-ESTRO-ESUR-ISUP-SIOG Guidelines on Prostate Cancer [10] the decision to carry out early diagnosis should be based on individual life expectancy, where comorbidity is at least as important as age, with the implementation of shared decision-making. However, until now, the available evidence has scarcely considered the presence of comorbidity when assessing the probability of overdiagnosis. The reason could be that particular comorbidities or a substantial number of comorbidities have, in some cases, led to the exclusion of patients from clinical trials due to heightened concerns regarding the risk of adverse events [28]. In addition, population-based studies do not usually consider patients' comorbidities when evaluate individual life expectancy.

Therefore, overdiagnosis in PCa screening can have significant implications for patient management and treatment decisions. When a diagnosis identifies an indolent or slow-growing cancer that may never cause symptoms or affect the patient's life expectancy, it often leads to unnecessary clinical interventions. These include frequent monitoring, invasive procedures or even treatments such as surgery or radiotherapy. These interventions can lead to adverse effects, such as incontinence and sexual dysfunction, with no clear benefit to the patient's survival or quality of life. We have shown in our study that older patients, or those with multiple comorbidities, are more likely to be overdiagnosed. This may mean that they are subjected to the psychological burden of a cancer diagnosis and the potential harms of overtreatment, rather than receiving the management of more relevant health problems. This underscores the need for a balanced approach to PSA testing, with clinicians carefully weighing the risks and

benefits and potentially opting for active surveillance or less aggressive interventions in patients with low-risk profiles. Personalising patient care in this way not only minimises the impact of overdiagnosis, but also aligns treatment decisions with each patient's overall health goals and quality of life.

## Limitations

Our study has several limitations. We used a previously described method to estimate the probability of overdiagnosis in our population. However, in clinical practice, there is no screened and unscreened population to estimate lead time, so we derived this estimation from the literature. This may have introduced potential biases by assuming uniform waiting times across diverse patient profiles. Consequently, the results may have limited external validity. Incorporating individualised waiting time estimate could have improved the precision and applicability of the results across broader demographic groups. Therefore, this approach is imperfect and may either overestimate the maximum lead time and underestimate overdiagnosis or underestimate the maximum lead time and overestimate overdiagnosis. Additionally, while we included patients' comorbidities to evaluate life expectancy, we limited this information to calculating the Charlson comorbidity index to assess the risk for each patient rather than examining specific comorbidities. Research on which comorbidities are linked to higher PSA levels and/or Gleason scores is needed.

Moreover, the rate of patients with an unknown Gleason score result was notable. However, we conducted a sensitivity analysis by assessing subgroups of patients with known Gleason scores. This analysis allowed for a more specific assessment of the likelihood of overdiagnosis in this subset. The results remained similar to those performed with the whole group. Lastly, the median follow-up was 5 years (IQR 3–8 years), which may be insufficient to observe long-term outcomes and accurately assess the true impact of screening on patient health. The low mortality rate during the follow-up period also makes it challenging to differentiate between cancers that would have progressed to cause harm and those that might have remained indolent or non-threatening if left untreated. A longer follow-up would allow a clearer assessment of the progression of indolent cancers, which could improve both the understanding of the risks of overdiagnosis and the applicability of findings to inform clinical decisions and long-term patient care strategies. These limitations highlight the complexity of evaluating overdiagnosis in clinical settings and emphasize the need for extended observation periods and comprehensive data collection to provide more reliable insights into the efficacy and consequences of screening programs. Finally, while the study provides valuable insights into the role of comorbidities in overdiagnosis in clinical practice, its results may not be fully applicable to other settings, especially those involving younger or healthier populations.

## Conclusions

In conclusion, our results indicated that PSA levels are significantly related to life expectancy at the time of diagnosis, considering both the patient's age and the number of comorbidities. This relationship highlights the crucial role of a patient's overall health status in determining the probability of overdiagnosis. Therefore, it is essential for healthcare providers to carefully consider a patient's comorbidities when deciding whether to order a new PSA test. This approach can help avoid unnecessary interventions and focus medical resources on patients who are more likely to benefit from screening, ultimately improving patient outcomes and reducing the risks associated with overdiagnosis. Hence, these results reinforce the importance of tailoring screening methods with more personalised screening strategies to optimise care and avoid overtreatment.

## Supporting information

**S1 Checklist. STROBE statement—Checklist of items that should be included in reports of cohort studies.**
(DOC)

## Author Contributions

**Conceptualization:** Abraham Beltrán, Lucy Parker, Irene Moral-Pérez, Juan Pablo Caballero-Romeu, Elisa Chilet-Rosell, Ildefonso Hernández-Aguado, Pablo Alonso-Coello, Elena Ronda, Luis Gómez-Pérez, Blanca Lumbreras.

**Formal analysis:** Abraham Beltrán.

**Methodology:** Abraham Beltrán, Blanca Lumbreras.

**Writing – original draft:** Abraham Beltrán, Blanca Lumbreras.

**Writing – review & editing:** Lucy Parker, Irene Moral-Pérez, Juan Pablo Caballero-Romeu, Elisa Chilet-Rosell, Ildefonso Hernández-Aguado, Pablo Alonso-Coello, Elena Ronda, Luis Gómez-Pérez.

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
