## [Decision Letter · Decision Letter 0]

6 Nov 2024

PONE-D-24-35992Impact of patients' age and comorbidities on prostate cancer overdiagnosis in clinical practicePLOS ONE

Dear Dr. Lumbreras,

Thank you for submitting your manuscript to PLOS ONE. After careful consideration, we feel that it has merit but does not fully meet PLOS ONE’s publication criteria as it currently stands. Therefore, we invite you to submit a revised version of the manuscript that addresses the points raised during the review process.

We look forward to receiving your revised manuscript.

Kind regards,

Prof. Jan Philipp Radtke, MBA

Academic Editor

PLOS ONE

**Journal Requirements:**

Research funded by the research project of the Instituto de Salud Carlos III, code PI20/01334, Principal Investigator Dr. Blanca Lumbreras, co-financed with FEDER funds from 631 the European Union “A way of doing Europe”. 

**Additional Editor Comments:**

General comments:

The study focuses on assessing the probability of overdiagnosis in the early detection of prostate cancer (PCa) using the PSA test. The study primarily investigates how the age of the patient and concomitant diseases affect overdiagnosis. The authors argue that most previous studies on PSA-based PCa screening have focussed on healthier populations, mainly from clinical trials, and want to provide a more relevant context by examining real-world clinical data. The aim of the study was to assess the likelihood of overdiagnosis based on PSA levels, Gleason scores and patients' comorbidities.

The study is based on a retrospective cohort analysis of 1,070 asymptomatic men over 40 who were diagnosed with PCa between 2004 and 2022. The primary outcome is the probability of overdiagnosis assessed by life expectancy adjusted for comorbidities using the Charlson Comorbidity Index. Lead times of 5, 10 and 15 years are considered in the study. Statistical comparisons are made between the groups stratified by PSA values and Gleason scores.

The paper contains several important results:

Higher PSA levels (>10 ng/dL) and Gleason scores (≥8) are associated with an increased likelihood of overdiagnosis, particularly in older patients with significant comorbidities.

Patients with lower PSA levels (4-10 ng/dL) and Gleason scores (≤7) were less likely to be overdiagnosed.

Radiation exposure was higher in patients with higher PSA levels, and pharmacological treatment was more common in patients with elevated PSA levels.

Most patients (71.7%) did not receive pharmacological treatment, highlighting the likelihood of overdiagnosis without active treatment.

In total, this is a retrospective observational study measure methods in a double center setting. Idea fine.

Specific comments:

1. Real-world data: The study’s use of real-world clinical data provides valuable insights, especially considering older and more comorbid patients who are often underrepresented in clinical trials. Very good.

2. Incorporation of comorbidities: The use of the Charlson Comorbidity Index to adjust life expectancy is a strong methodological approach, as it emphasizes the importance of considering a patient’s overall health, not just age. Well, done.

3. Comprehensive analysis: The inclusion of different lead times (5, 10, and 15 years) provides a thorough examination of overdiagnosis probabilities. Well, done

4. Follow-up duration: The median follow-up of 5.7 years, though reasonable, may not be sufficient to capture long-term outcomes or fully assess the impact of overdiagnosis on patient health. � Extension of the follow-up if necessary

5. Unknown Gleason scores: A considerable portion of patients did not have their Gleason scores recorded, which could have affected the robustness of the statistical comparisons. � severely limits the significance of the study

6. Overdiagnosis estimation: The estimation of overdiagnosis relies on external lead-time data from the literature rather than patient-specific lead-time information, which introduces potential biases.The study's findings may be less applicable to younger or healthier populations, limiting the external validity.

The study emphasises the importance of considering both age and comorbidities when assessing the risk of overdiagnosis in prostate cancer screening. The study shows that older patients with significant comorbidities are more likely to be overdiagnosed, often without receiving active treatment. The results underline the need for personalised screening strategies and caution in the general use of PSA testing without consideration of patients' individual health profiles.

The study addresses an important issue in PCa screening and provides useful clinical insights. The limitations regarding the data on Gleason scores and follow-up duration should be re-examined as they significantly limit the statistical power. The paper could also benefit from further clarification of the impact of overdiagnosis on patient management and treatment decisions. Overall, the research findings are solid and relevant, especially for clinicians looking to adjust PSA testing strategies based on individual patient risk factors.

Reviewers' comments:

Reviewer's Responses to Questions

**Comments to the Author**

1. Is the manuscript technically sound, and do the data support the conclusions?

Reviewer #1: Yes

2. Has the statistical analysis been performed appropriately and rigorously? 

Reviewer #1: Yes

3. Have the authors made all data underlying the findings in their manuscript fully available?

Reviewer #1: Yes

4. Is the manuscript presented in an intelligible fashion and written in standard English?

Reviewer #1: Yes

5. Review Comments to the Author

**Reviewer #1:** General comments:

The study focuses on assessing the probability of overdiagnosis in the early detection of prostate cancer (PCa) using the PSA test. The study primarily investigates how the age of the patient and concomitant diseases affect overdiagnosis. The authors argue that most previous studies on PSA-based PCa screening have focussed on healthier populations, mainly from clinical trials, and want to provide a more relevant context by examining real-world clinical data. The aim of the study was to assess the likelihood of overdiagnosis based on PSA levels, Gleason scores and patients' comorbidities.

The study is based on a retrospective cohort analysis of 1,070 asymptomatic men over 40 who were diagnosed with PCa between 2004 and 2022. The primary outcome is the probability of overdiagnosis assessed by life expectancy adjusted for comorbidities using the Charlson Comorbidity Index. Lead times of 5, 10 and 15 years are considered in the study. Statistical comparisons are made between the groups stratified by PSA values and Gleason scores.

The paper contains several important results:

Higher PSA levels (>10 ng/dL) and Gleason scores (≥8) are associated with an increased likelihood of overdiagnosis, particularly in older patients with significant comorbidities.

Patients with lower PSA levels (4-10 ng/dL) and Gleason scores (≤7) were less likely to be overdiagnosed.

Radiation exposure was higher in patients with higher PSA levels, and pharmacological treatment was more common in patients with elevated PSA levels.

Most patients (71.7%) did not receive pharmacological treatment, highlighting the likelihood of overdiagnosis without active treatment.

In total, this is a retrospective observational study measure methods in a double center setting. Idea fine.

Specific comments:

1. Real-world data: The study’s use of real-world clinical data provides valuable insights, especially considering older and more comorbid patients who are often underrepresented in clinical trials. Very good.

2. Incorporation of comorbidities: The use of the Charlson Comorbidity Index to adjust life expectancy is a strong methodological approach, as it emphasizes the importance of considering a patient’s overall health, not just age. Well, done.

3. Comprehensive analysis: The inclusion of different lead times (5, 10, and 15 years) provides a thorough examination of overdiagnosis probabilities. Well, done

4. Follow-up duration: The median follow-up of 5.7 years, though reasonable, may not be sufficient to capture long-term outcomes or fully assess the impact of overdiagnosis on patient health. � Extension of the follow-up if necessary

5. Unknown Gleason scores: A considerable portion of patients did not have their Gleason scores recorded, which could have affected the robustness of the statistical comparisons. � severely limits the significance of the study

6. Overdiagnosis estimation: The estimation of overdiagnosis relies on external lead-time data from the literature rather than patient-specific lead-time information, which introduces potential biases.The study's findings may be less applicable to younger or healthier populations, limiting the external validity.

The study emphasises the importance of considering both age and comorbidities when assessing the risk of overdiagnosis in prostate cancer screening. The study shows that older patients with significant comorbidities are more likely to be overdiagnosed, often without receiving active treatment. The results underline the need for personalised screening strategies and caution in the general use of PSA testing without consideration of patients' individual health profiles.

The study addresses an important issue in PCa screening and provides useful clinical insights. The limitations regarding the data on Gleason scores and follow-up duration should be re-examined as they significantly limit the statistical power. The paper could also benefit from further clarification of the impact of overdiagnosis on patient management and treatment decisions. Overall, the research findings are solid and relevant, especially for clinicians looking to adjust PSA testing strategies based on individual patient risk factors.

6. PLOS authors have the option to publish the peer review history of their article (what does this mean?). If published, this will include your full peer review and any attached files.

Reviewer #1: No

---

## [Author Response · Author response to Decision Letter 0]

18 Nov 2024

Journal Requirements:

1. When submitting your revision, we need you to address these additional requirements. Please ensure that your manuscript meets PLOS ONE's style requirements, including those for file naming. The PLOS ONE style templates can be found at 

Done

Research funded by the research project of the Instituto de Salud Carlos III, code PI20/01334, Principal Investigator Dr. Blanca Lumbreras, co-financed with FEDER funds from 631 the European Union “A way of doing Europe”. 

Done

We have added this statement in the manuscript and in the system:

We will make our data fully accessible in a public repository upon acceptance of this manuscript (https://fairsharing.org/), in accordance with PLOS ONE’s open access requirements. We understand that all data, except data restricted by ethics protocol compliance, must be freely accessible upon publication. 

With this expression we wanted to express that there was no table related to the results. We did not mean that there were data not shown in the article (this was an error). All results described in the text are reflected in figure 3 (figure 3A and figure 3B), so we have deleted this expression and rearranged the results (page 18). For the results on page 19, all data are described in the text.

Additional Editor Comments:

General comments:

The study focuses on assessing the probability of overdiagnosis in the early detection of prostate cancer (PCa) using the PSA test. The study primarily investigates how the age of the patient and concomitant diseases affect overdiagnosis. The authors argue that most previous studies on PSA-based PCa screening have focussed on healthier populations, mainly from clinical trials, and want to provide a more relevant context by examining real-world clinical data. The aim of the study was to assess the likelihood of overdiagnosis based on PSA levels, Gleason scores and patients' comorbidities.

The study is based on a retrospective cohort analysis of 1,070 asymptomatic men over 40 who were diagnosed with PCa between 2004 and 2022. The primary outcome is the probability of overdiagnosis assessed by life expectancy adjusted for comorbidities using the Charlson Comorbidity Index. Lead times of 5, 10 and 15 years are considered in the study. Statistical comparisons are made between the groups stratified by PSA values and Gleason scores.

The paper contains several important results: Higher PSA levels (>10 ng/dL) and Gleason scores (≥8) are associated with an increased likelihood of overdiagnosis, particularly in older patients with significant comorbidities.

Patients with lower PSA levels (4-10 ng/dL) and Gleason scores (≤7) were less likely to be overdiagnosed. Radiation exposure was higher in patients with higher PSA levels, and pharmacological treatment was more common in patients with elevated PSA levels. Most patients (71.7%) did not receive pharmacological treatment, highlighting the likelihood of overdiagnosis without active treatment. In total, this is a retrospective observational study measure methods in a double center setting. Idea fine.

Specific comments:

1. Real-world data: The study’s use of real-world clinical data provides valuable insights, especially considering older and more comorbid patients who are often underrepresented in clinical trials. Very good.

Thank you very much for your comment. Indeed, the use of real-world clinical data in the study allows us to capture perspectives that are often not considered in controlled clinical trials, especially in populations such as older adults or people with multiple conditions.

2. Incorporation of comorbidities: The use of the Charlson Comorbidity Index to adjust life expectancy is a strong methodological approach, as it emphasizes the importance of considering a patient’s overall health, not just age. Well, done.

Thank you for your comment. This approach recognises that a patient's comorbidities can significantly influence his life expectancy and, consequently, the likelihood of overdiagnosis.

3. Comprehensive analysis: The inclusion of different lead times (5, 10, and 15 years) provides a thorough examination of overdiagnosis probabilities. Well done.

Thank you very much. The use of multiple time periods (5, 10 and 15 years) in the study allows for a more in-depth analysis of the probabilities of overdiagnosis, providing information on how the probability of overdiagnosis changes over time.

4. Follow-up duration: The median follow-up of 5.7 years, though reasonable, may not be sufficient to capture long-term outcomes or fully assess the impact of overdiagnosis on patient health. � Extension of the follow-up if necessary

As noted in the limitations section, the median follow-up duration of 5.7 years provides valuable information, but may be insufficient for a full analysis of the long-term effects of overdiagnosis on patient health and mortality. We have included in the limitation section the following sentence:

‘A longer follow-up would allow a clearer assessment of the progression of indolent cancers, which could improve both the understanding of the risks of overdiagnosis and the applicability of findings to inform clinical decisions and long-term patient care strategies’ (page 24, lines 654-657).

5. Unknown Gleason scores: A considerable portion of patients did not have their Gleason scores recorded, which could have affected the robustness of the statistical comparisons. � severely limits the significance of the study

The absence of recorded Gleason scores for a significant number of patients is a limitation that could have affected the robustness and significance of the study results. This data gap highlights the challenge of conducting analyses in real-world practice where not all variables are consistently documented, and this could limit the applicability of the study to larger clinical populations. Nevertheless, we conducted a sensitivity analysis by assessing subgroups of patients with known Gleason scores. This analysis allowed for a more specific assessment of the likelihood of overdiagnosis in this subset. The results remained similar to those performed with the whole group. We have added this explanation in the limitation section (page 24, lines 646-649).

6. Overdiagnosis estimation: The estimation of overdiagnosis relies on external lead-time data from the literature rather than patient-specific lead-time information, which introduces potential biases. The study's findings may be less applicable to younger or healthier populations, limiting the external validity.

As discussed in the study limitations, the study's approach to estimating overdiagnosis uses waiting time data from the literature, rather than patient-specific waiting times. This may have introduced potential biases by assuming uniform waiting times across diverse patient profiles. Consequently, the results may have limited external validity. Incorporating individualised waiting time estimate could improve the precision and applicability of the results across broader demographic groups. We have expanded the limitation in the discussion section to clarify this point (pages 23-24, lines 634-638).

The study emphasises the importance of considering both age and comorbidities when assessing the risk of overdiagnosis in prostate cancer screening. The study shows that older patients with significant comorbidities are more likely to be overdiagnosed, often without receiving active treatment. The results underline the need for personalised screening strategies and caution in the general use of PSA testing without consideration of patients' individual health profiles.

The reviewer is right. The study highlights the critical role of assessing both age and comorbidities to more accurately gauge the risk of overdiagnosis in prostate cancer screening. By demonstrating that older patients with significant comorbidities are more likely to be overdiagnosed - often leading to diagnoses that do not result in active treatment - the results reinforce the importance of tailoring screening methods with more personalised screening strategies to optimise care and avoid overtreatment. We have remarked this idea in the conclusion section (page 25, lines 677-679).

The study addresses an important issue in PCa screening and provides useful clinical insights. The limitations regarding the data on Gleason scores and follow-up duration should be re-examined as they significantly limit the statistical power. The paper could also benefit from further clarification of the impact of overdiagnosis on patient management and treatment decisions. Overall, the research findings are solid and relevant, especially for clinicians looking to adjust PSA testing strategies based on individual patient risk factors.

The reviewer is right. Limitations related to missing Gleason scores and the relatively short duration of follow-up warrant further attention, as they limit statistical power and the ability to observe long-term results. As we mentioned previously, we have extended the explanation for both limitations in the discussion section to further characterise their potential impact on the study results. 

In addition, we have included a paragraph to detail the impact of overdiagnosis on patient management and treatment decisions in the Implication for clinical practice section (page 23, lines 613-628):

‘Overdiagnosis in PCa screening can have significant implications for patient management and treatment decisions. When a diagnosis identifies an indolent or slow-growing cancer that may never cause symptoms or affect the patient's life expectancy, it often leads to unnecessary clinical interventions. These include frequent monitoring, invasive procedures or even treatments such as surgery or radiotherapy. These interventions can lead to adverse effects, such as incontinence and sexual dysfunction, with no clear benefit to the patient's survival or quality of life. We have shown in our study that older patients, or those with multiple comorbidities, are more likely to be overdiagnosed. This may mean that they are subjected to the psychological burden of a cancer diagnosis and the potential harms of overtreatment, rather than receiving the management of more relevant health problems. This underscores the need for a balanced approach to PSA testing, with clinicians carefully weighing the risks and benefits and potentially opting for active surveillance or less aggressive interventions in patients with low-risk profiles. Personalising patient care in this way not only minimises the impact of overdiagnosis, but also aligns treatment decisions with each patient's overall health goals and quality of life’.

---

## [Editor Report · Decision Letter 1]

4 Dec 2024

Impact of patients' age and comorbidities on prostate cancer overdiagnosis in clinical practice

PONE-D-24-35992R1

Dear Dr. Lumbreras,

We’re pleased to inform you that your manuscript has been judged scientifically suitable for publication and will be formally accepted for publication once it meets all outstanding technical requirements.

Kind regards,

Jan Philipp Radtke, MBA

Academic Editor

PLOS ONE
---

## [Editor Report · Acceptance letter]

21 Dec 2024

PONE-D-24-35992R1 

PLOS ONE

Dear Dr. Lumbreras, 

I'm pleased to inform you that your manuscript has been deemed suitable for publication in PLOS ONE. Congratulations! Your manuscript is now being handed over to our production team.

Kind regards, 

on behalf of

Professor Dr. Jan Philipp Radtke 

Academic Editor

PLOS ONE